# Iterative Learning-Based Path and Speed Profile Optimization for an Unmanned Surface Vehicle

**DOI:** 10.3390/s20020439

**Published:** 2020-01-13

**Authors:** Yang Yang, Quan Li, Junnan Zhang, Yangmin Xie

**Affiliations:** 1Research Institute of USV Engineering, Shanghai University, Shanghai 200444, China; yangyang_shu@shu.edu.cn (Y.Y.); shu_liquan@shu.edu.cn (Q.L.); zjnshu@shu.edu.cn (J.Z.); 2Shanghai Key Laboratory of Intelligent Manufacturing and Robotics, Shanghai University, Shanghai 200444, China

**Keywords:** USV, iterative parameter-tuning, path-smoothing, speed profile design

## Abstract

Most path-planning algorithms can generate a reasonable path by considering the kinematic characteristics of the vehicles and the obstacles in hydrographic survey activities. However, few studies consider the influence of vehicle dynamics, although excluding system dynamics may considerably damage the measurement accuracy especially when turning at high speed. In this study, an adaptive iterative learning algorithm is proposed to optimize the turning parameters, which accounts for the dynamic characteristics of unmanned surface vehicles (USVs). The resulting optimal turning radius and speed are used to generate the path and speed profiles. The simulation results show that the proposed path-smoothing and speed profile design algorithms can largely increase the path-following performance, which potentially can help to improve the measurement accuracy of various activities.

## 1. Introduction

Unmanned surface vehicles (USVs), also called autonomous surface craft (ASC), are self-driving marine vessels widely used in environmental monitoring and hydrographic survey [1]. Path-following performance of the USVs is one of the main factors influencing the measurement accuracy in hydrographic survey activities [2].

A reliable autonomous navigation system with the capabilities of path-planning, obstacle avoidance, and auto-guidance is essential for an USV to deal with various and dynamic marine situations [3]. The most widely used obstacle avoidance and path-planning algorithms include artificial potential fields (APF) [4], rapidly exploring random trees (RRTs) [5], greedy mechanism-based particle swarm optimization (PSO) [6], and grid map-based path-planning algorithms (e.g., A* algorithm [7], Field D* [8], and Theta* [9]). These algorithms focus on path-planning for obstacle avoidance considering the kinematics of the vehicle and the obstacles.

By using the abovementioned techniques, a segmented, first-order discontinuous line-shape path is obtained, as shown in Figure 1a. The discontinuity does not considerably affect the path-following performance when the USV drives at a relatively low speed but can induce large differences between the planned and achieved paths when the speed is high and the dynamics of the vehicle become non-neglectable. The deviation causes the vehicle to fail the obstacle avoidance task when the vehicle is required to make sharp turns for non-holonomic systems, such as USVs [10,11] and, thus, should be diminished. Typically, there are two solutions. The first solution is to improve the mechanical and/or control system so that the vehicle itself could be sufficiently agile to make accurate and shape turns. The second solution is to modify the path and the speed profiles in Figure 1b so that they are more suitable for the USV to follow. In Figure 1b, the path is bent to arcs at the sharp corners and better fits the dynamic characteristics of the USV.

In this study, we focus on the work of the latter solution. Specifically, by assuming that the physical system and motion control of the USV is already provided, we study how the path and speed profiles can be optimized so that the overall performance of path-following and obstacle avoidance can be improved. There are usually two approaches to solve such a motion planning problem [12]: (1) trajectory planning with a simultaneous optimization of path and speed profiles; (2) geometric path-planning followed by velocity planning, which is also known as path-velocity decomposition. The latter is easier to implement in real time where path-planning usually considers geometric continuity and velocity planning addresses the kinematic constraints.

To achieve path smoothness, the Dubins curve [13] is a good choice. It is created by concatenating line segments and arcs of circles. The Dubins curve is first-order continuous and thus smoother than the original line-based path. In addition, it is computationally cost efficient, which makes it suitable for real-time path-planning cases. The maximum curvature, or correspondingly, the minimum turning radius of the system, is used as a constraint to generate the optimal Dubins curve [14,15]. To some extent, this approach considers the dynamic properties of the vehicle. However, because the minimum turning radius is closely related to the vehicle speed, the isolation of path and speed profile design may compromise the overall path-following performance. There are other studies that use second-order continuous paths such as cubic spirals [16], Bezier splines [10], clothoidal spirals [17,18], quintic polynomials [19], polar splines [20] and cardioids [21]. These approaches may further smooth the path but also make the computation more complex and time consuming. Therefore, the decision of whether to adopt first-order or second-order curves should be balanced by considering the mission demands on path continuity and real-time ability.

For speed-planning strategies, researchers usually set up a time-optimal problem with constraints from the kinematic properties of the vehicles including maximum speed, longitudinal and lateral acceleration and jerk [22]. The kinematic model of the vehicle is established, and the key design parameters regarding the velocity profile are specified to form a numerical optimization problem [23], the solution of which defines the final speed profile. In some cases, the experimental choice of the speed is used to define a speed profile when the time-optimality is not an essential requirement [24,25]. There are also studies that take into consideration the kinematic feasibility, and model predictive control is introduced to generate a speed profile based on the predicted performance [18].

However, few studies mention the consideration of vehicle dynamics for either path-planning or speed-planning though the exclusion of the dynamics of the system may largely compromise the path tracking performance especially at high speed turning. There are mainly two reasons for this. First, the use of a dynamic model may increase the complexity in the formulation and computation of the optimization problem. Second, in most cases, an accurate dynamical model is not easy to obtain. Therefore, this study realizes a practical and real-time solution for the path and speed profile generation by considering the influence of vehicle dynamics on path-following performance. To achieve this goal, the schematic of the path and speed planner is shown in Figure 2. The offline learning block uses experimental data to obtain an optimal turning radius and speed pair for the vehicle, which provides design constraints for the smooth path-following and speed profile optimization. Thus, the generated path and speed profile are customized for the specific vehicle according to its dynamical properties.

In summary, previous studies usually emphasized the use of methods that generate smooth and optimal path or/and speed profiles according to their geometric or kinematic properties. However, this study provides a practical and experiment-based approach that takes the USV dynamics into consideration by:Determining an optimal arc radius and the corresponding turning speed using an offline iteratively tuning method.Generating a smooth trajectory with a provided arc radius and ensuring small path deviation from the original line segments.Planning optimal speed profile on the line-arc combined trajectory using step response curves of the USV.

The remainder of the paper is structured as follows. Section 2 presents the iterative tuning method and its application in determining an optimal arc radius and corresponding turning speed. Then, the method of path-smoothing with fixed turning radius is introduced in Section 3, followed by an optimal speed-planning strategy in Section 4. Section 5 shows the simulation verification. Finally, Section 6 presents concluding remarks.

## 2. Iterative Learning Method for Path Parameter-Tuning

### 2.1. Problem Setup

Iterative learning is broadly used to deal with repetitive tasks by updating control signals or system parameters. It learns the system behavior via repeated tests which is possible to achieve optimal performance without providing an explicit system model [24]. Iterative learning sometimes is called a data-driven approach [26]. When the control signals are updated iteration by iteration, it is called iterative learning control (ILC) [27]; if the parameters in the systems are updated during the process, it is called iterative parameter-tuning (IPT) [28]. In this paper, IPT is used to obtain a suitable combination of the references of the turning radius and speed for a specific USV to decrease the path-following error with limited sacrifice of turning speed and turning radius. Generally speaking, small turning radius and large turning speed induce large path deviation. With proper selection on the weight factors for the three aforementioned terms, IPT can provide an optimized solution with balanced path-following performance.

The parameter vector used for tuning is $x=\left(r_{r},v_{r}\right)$, where $r_{r}$ is the reference of the turning radius and $v_{r}$ is the reference of the forwarding speed at the turning corner. The cost function is defined as
(1)$$J\left(x\right)=\lambda_{1}{\left(r_{r}-r_{\min}\right)}^{2}+\lambda_{2}{\left(v_{r}-v_{d}\right)}^{2}+\lambda_{3}\frac{1}{n}\sum_{i=1}^{n}∥{\bar{p}}_{i}-p_{i}∥$$
where $r_{\min}$ is the empirical value that defines the minimum turning radius of USV when it is within its normal speed range; $v_{d}$ is also provided by the user and specifies the desired speed that the USV will drive at, which is generally chosen as a relatively high speed value to speed up the turning action; $p_{i}$ is the reference trajectory vector generated by $r_{r}$ and $v_{r}$; ${\bar{p}}_{i}$ is the actual trajectory vector; and $\lambda_{1}$, $\lambda_{2}$ and $\lambda_{3}$ are the weight factors. The first term in *J* represents the difference between the reference radius with the minimum turning radius; the second term evaluates the difference between the reference speed with the desired speed; and the last terms show the tracking error under the current setup of $r_{r}$ and $v_{r}$. By choosing different $\lambda_{1}$, $\lambda_{2}$ and $\lambda_{3}$, the user can specify the respective emphasis on the three abovementioned aspects.

### 2.2. Two-Point Step-Size Adaptation (TPA)-IPT Algorithm

The goal of IPT is to obtain the smallest *J* by iteratively tuning the parameter of $r_{r}$ and $v_{r}$, which can be treated as a constrained optimization problem below. Similar to $r_{\min}$, $r_{\max}$, $v_{\min}$ and $v_{\max}$ are also user-specified parameters based on the USV system capabilities.
(2)$$\begin{array}{l} \min J\left(r_{r},v_{r}\right)\\ r_{\min}<r_{r}<r_{\max}\\ v_{\min}<v_{r}<v_{\max} \end{array}$$

In this study, the line search method schematic [29] is adopted to identify an optimal *J*. The updating law of the path parameter is written as
(3)$$x_{k+1}=x_{k}+\alpha_{k}p_{k}$$
where $\alpha_{k}$ is the step length, and $p_{k}$ is the search direction. In general, $p_{k}$ is a descent direction and frequently has the form of
(4)$$p_{k}=-B_{k}^{-1}\nabla J_{k}$$
where $B_{k}^{-1}$ is the symmetric and nonsingular matrix, and $\nabla J_{k}$ is the gradient. When ${B_{k}}^{-1}=I$, where I is the identity matrix, it is the fastest descent direction. The selection of the step length $\alpha_{k}$ is highly related to the gradient $\nabla J_{k}$, and these parameters must satisfy $|1-\alpha_{k}\nabla J_{k}^{T}|\leq \rho <1$ to guarantee stability, where ρ is a positive scaler. If $\nabla J_{k}$ is known, the updating gains can be chosen as $p_{k}=-\nabla J_{k}$ and $\alpha_{k}=\nabla {J_{k}}^{-1}$. In reality, *J* is an unknown nonlinear function of x. Therefore, an additional algorithm needs to be developed to identify steady descent directions and step length, which is not a trivial task.

Particularly for the tuning task in this study, the dimension of the variable space is as low as 2, which make it possible to estimate a descent direction with a limited number of tests. Taking the $k^{th}$ iteration as an example, the procedure starts at a given point x. Four trials provide an estimation of the gradient direction at x by
(5)$$y_{kj}=x_{k}+\Delta_{kj}$$
where *j* is the trial index number with $\Delta_{k1}=\left(+\mu \alpha_{k},0\right),\Delta_{k2}=\left(-\mu \alpha_{k},0\right),\Delta_{k3}=\left(0,+\alpha_{k}\right),\Delta_{k4}=\left(0,-\alpha_{k}\right)$. Where $\mu =\frac{r_{\max}-r_{\min}}{v_{\max}-v_{\min}}$ is the normalized coefficient. $\alpha_{k}$ represents the step size of the $(k{-1)}^{th}$ parameter update, and it is the step length of the previous iterations for the purpose of saving parameter-tuning trials. The calculation of an adaptive step length will be further explained later. Based on the test results, the cost function values can be obtained, and the unit gradient vector $e_{k}$ is numerically estimated from the previous two iterations by using Equations (Equation 6) and (Equation 7).
(6)$$D_{k}=\sum_{j=1}^{4}\left|J\left(y_{j}\right)-J\left(x_{k}\right)\right|\frac{J\left(y_{j}\right)-J\left(x_{k}\right)}{y_{j}-x_{k}}$$
(7)$$e_{k}=\frac{D_{k}}{∥D_{k}∥}$$

The second step to update x is to determine the line search step length. Considering the cost of conducting many tests, it is essential for the tuning process to converge as fast as possible to be applicable to real systems. Therefore, a large step size is preferred for the IPT algorithm to speed up the convergence and reduce the needed number of trials. However, convergence occurs only if the step length is sufficiently small [30]. Therefore, a rule that can dynamically adapt the step size among iterations needs to be developed.

Self-adaptive algorithms for the step length are not a new concept in numerical optimization. A typical method is called Two-Point Step-Size Adaptation (TPA), where only two trial points are needed to determine the change of the step length between iterations. This method works well with the general optimization setups such as the steepest descent method [30] and the evolutionary gradient search [31]. Another method combines the adjustment strategy of the step length with the evolution optimization method [32], where the step length is calculated by the weighted distance sum of randomly generated offspring with descending costs. To make the algorithm robust enough, a sufficient number of offspring is required. A cumulative step-size adaptation (CSA) method is further developed to better remove the randomness of evolution strategies [33]. The step length is determined by historical records and is optimized accordingly. However, the authors admit its sensitivity to the sampling procedure and replace it with TPA in their future work [34].

Based on the short review of the abovementioned self-adaptive step length strategies, it is clear that TPA stands out for its simplicity and widely proven robustness. TPA is suitable for the IPT framework because only two trials are required for the step length adaptation in each iteration. Therefore, in this study, we propose to merge the TPA method with the tuning framework to obtain the TPA-IPT algorithm, as shown in Algorithm 1. In addition, the tuned step-size updates the parameters and functions as the off-distance of the next trial points, which is similar to [32].

For each iteration, extra two trials are proceeded by two different step lengths, $\alpha_{k}\zeta$ and $\alpha_{k}/\zeta$, where ζ∈(0,1) is a user-tuned number. The comparison of the two trials in terms of their cost function values provides a superior choice between the two for the step-size update, as shown in (Equation 8).
(8)αk+1=αkζ,J(xk−αkζek)≤J(xk−αkαkζζek)αkαkζ,J(xk−αkζek)≥J(xk−αkαkζζek)ζ,J(xk−αkζek)≥J(xk−αkαkζζek)

Thus, the updating rule of the path parameters is obtained as
(9)$$x_{k+1}=x_{k}-\alpha_{k+1}e_{k}$$

When the cost function value starts to increase as $J\left(x_{k+1}\right)\geq J\left(x_{k}\right)$, the variables are close to the minimum point. Then, the abovementioned TPA rule can induce vibrations around the minimum and slow down the convergence. Therefore, the dichotomy is adopted to quickly decrease the step size by imposing ${\alpha^{*}}_{k+1}=\frac{1}{2^{n}}\alpha_{k}$, where *n* is the smallest positive integer that satisfies this condition $J\left({x^{*}}_{k+1}\right)<J\left(x_{k}\right)$.

The abovementioned optimization process is terminated when $\left|{x^{*}}_{k+1}-x_{k}\right|\leq \psi$, where ψ is a user-specified positive value.

**Algorithm 1:** TPA-IPT.

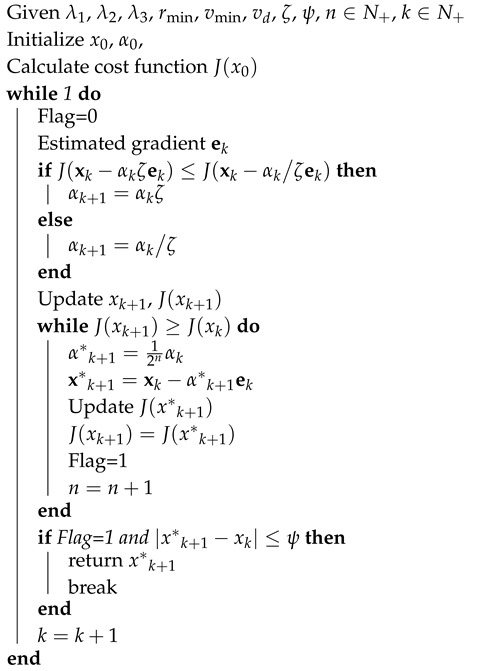



## 3. Path-Smoothing Algorithm

The paths generated by the path-planning algorithm usually consist of line sets connected by first-order discontinuous corners. In general, due to their dynamics at high speed, the USV systems cannot follow the path accurately at the corners, and the offset of the real trajectory from the planned one sometimes causes problems such as obstacle collision or undesirable path-following performance. To modify the original non-smooth path to be more suitable for a USV to follow, the optimized tuning radius determined by the IPT algorithm in Section 2 is used for path-smoothing in this section.

The path-smoothing problem is set up as follows. Given the start point $w_{1}$ and the target point $w_{3}$, the path-smoothing problem is defined to find a path *w* such that:*w* is based on the original line-segment path and satisfies the dynamic characteristics and security requirements such as the curvature (turning radius) limitation.*w* begins at the starting point $w_{1}$ and ends at the target point $w_{3}$.*w* does not collide with the obstacles.The path deviation De between the smoothed path and the original path should be small.

The Dubins curves are adopted to smooth the path with slight improvement compared to the previously used approaches since they require a low computation cost and are easy to apply. In a previous study, typically one arc or two arcs are used to smooth the corners [14,15,35], as shown in Figure 3. The black solid lines represent the original path, and the red solid lines represent the smoothed path. Break points usually imply nearby obstacles, and the deviation from the original path increases the risk of collisions between obstacles and the smoothed path.

In this study, to decrease the path deviation and alleviate the collision risk, we propose to use three Dubins curves for every break point, which is named the L-3C-L method, as shown in Figure 4. $w_{1}$, $w_{2}$ and $w_{3}$ are the path points on the original path, and $w_{1},p_{1},p_{2},p_{3},p_{4}$ and $w_{3}$ are the path points on the smoothed path. $O_{2}$ is in the angle bisector of the path angle.

To compare the path deviation using the methods shown in Figure 3 and the L-3C-L method, their theoretical formulas are provided below.

The path deviation [13] of the method in Figure 3a is:(10)$$De_{a}=r\left(\frac{1}{\sin\frac{\theta }{2}}-1\right)$$
where *r* is the turning radius obtained by the IPT algorithm, and θ is the corner angle of the original path.

The path deviation [35] of the method in Figure 3b is:(11)$$De_{b}=r\left(1+\cos\theta \right)$$

The path deviation of the L-3C-L method is:(12)$$De_{L3CL}=r\left(1-\sin\frac{\theta }{2}\right)$$

Figure 5 shows the deviation comparison of the three smoothing methods: one arc, two arcs, and L-3C-L methods, where De is the largest path deviation, and *r* is the radius of the corner. The path deviation of the L-3C-L algorithm is the smallest of the three smoothing algorithms, even at very sharp corners.

There is a special case when two corners are very close to each other, as shown in Figure 6. $w_{1}$, $w_{2}$, $w_{3}$ and $w_{4}$ represent the original path, and $w_{1},p_{1},p_{2},p_{3},p_{4},p_{5},p_{6}$ and $w_{4}$ represent the smoothed path using our method. $p_{3}p_{4}$ is the common tangent of the circle $O_{2}$ and $O_{3}$. In this case, two L-3C-L corners are merged to be two L-2C-L corners. The corresponding two deviations are:(13)$$\left\{\begin{array}{l} De_{1}=r\left(1-\sin\frac{\theta_{1}}{2}\right)\\ De_{2}=r\left(1-\sin\frac{\theta_{2}}{2}\right) \end{array}\right.$$

## 4. Optimal Speed Profile Design

Using the path-planning algorithm proposed in Section 3, any path can be decomposed into a succession of turns composed of circular arcs and straight lines. This is very useful in finding closed-form optimal speed profiles because both straight-line segments and circle arcs can be associated with constant speeds. The IPT algorithm in Section 2 provides an optimized turning speed, which differs from the desired USV speed when it follows straight lines. Therefore, when a turn is initiated and terminated, the velocity of the system should be regulated so that it can smoothly switch the motion status between the two modes.

Therefore, in this study, a method is proposed to plan the velocity. The work is simplified to plan the process of accelerations and decelerations between the two turns because the turning speed and the line-following speed have both been fixed. Thus, the overall speed profile will consist of three families of curves.
Constant speed curves at a minimum value $v_{r}$ when the curvature profile is a circular arc. $v_{r}$ is obtained from Section 2.Constant cruising speed at the straight line with a cruising speed of the USV denoted as $v_{c}$.Changing speed profile between $v_{r}$ and $v_{c}$ when the USV switches between the turning mode and the line-following mode. For these stages, there are two conditions to consider.

Condition 1: If the straight-line segment is sufficiently long, the USV will perform a full acceleration and deceleration at the two ends of the line, as shown in Figure 7a. The corresponding minimum length of the line segment is denoted as $d_{\min}$.

Condition 2: If the straight-line segment between two turns is not sufficiently long, $d<d_{\min}$, the USV will not be able to accelerate to $v_{c}$. Instead, it decelerates to some middle point to make sure that the velocity is $v_{r}$ at the next turn, as shown in Figure 7b.

To determine whether the straight-line segment between two turns is sufficiently long for a specific USV, a test is needed to estimate the minimum length of the line segment. The path length of the acceleration section $d_{a}$ and of the deceleration section $d_{d}$ are recorded when the USV follows the given command. For a USV without a closed-loop velocity control, a full-power acceleration/deceleration test is required. When the USV has a velocity control unit, the command is usually given as an expected velocity as $v_{c}$ when accelerating and $v_{r}$ when decelerating, as shown in Figure 8a. The minimum length of the line segment for condition 1 is $d_{\min}=d_{a}+d_{d}$, as shown in Figure 8b.

For the first condition $d\geq d_{\min}$ in Figure 7a, the USV can perform a full acceleration and deceleration at the two ends of the line. The acceleration and deceleration positions in Figure 7a are chosen to make $P_{1}P_{2}=d_{a}$ and $P_{3}P_{4}=d_{d}$.

For the second condition $d<d_{\min}$ in Figure 7b, we need further treatment to find the switch point $P_{2}$ from acceleration to deceleration. The functions for the acceleration and deceleration curves are denoted as $f_{1}\left(t\right)$ and $f_{2}\left(t\right)$, respectively. Assuming that the USV requires $t_{x1}$ to accelerate and $t_{x2}$ to decelerate so that it has speed $v_{r}$ at $p_{3}$, the equations in Equation (Equation 14) should be satisfied. $t_{x1}$ and $t_{x2}$ can be solved, and the switching point $p_{2}$ is chosen to make $P_{1}P_{2}=\int_{0}^{t_{x1}}f_{1}\left(t\right)dt$.
(14)$$\left\{\begin{array}{c} \int_{0}^{t_{x1}}f_{1}\left(t\right)dt+\int_{t_{d}-t_{x2}}^{t_{d}}f_{2}\left(t\right)dt=d\\ f_{1}\left(t_{x1}\right)=f_{2}\left(t_{d}-t_{x2}\right) \end{array}\right.$$

## 5. Simulation Results and Discussion

Since most ships do not have actuators in all degrees of freedom, the reduced-order model is often used. The 3 DOF horizontal plane model (surge, sway, yaw) is widely used in trajectory tracking control systems [36] with the assumption of planar motion. A classic 3 DOF USV model, the accuracy of which is verified in [37], is adopted in the following simulation work. The parameters of the vessel system are shown in Table 1 and Table 2. Additional details on the dynamic model are presented in Appendix A, and the readers are referred to [37] for further information.

### 5.1. IPT Results

According to the literature [37], the maximum speed $v_{\max}$ of the USV is 18 kn. By using empirically determined values, the minimum turning radius, minimum speed and desired turning velocity are set as $r_{\min}=20$ m, $v_{\min}=1$ kn and $v_{d}=v_{\max}=18$ kn. In the cost function, $\lambda_{1}$, $\lambda_{2}$ and $\lambda_{3}$ emphasize the turning radius, turning speed and path deviation, respectively. Users can tune the parameters accordingly to specific task demands. The weights of the cost function in this paper are chosen as $\left(\lambda_{1},\lambda_{2},\lambda_{3}\right)=\left(1,3.05,278\right)$ to emphasize on decreasing path deviations at turning corners rather than the needs to obtain small turning radius and high turning speed. The initial step size and step-size adjustment coefficient are set to $\alpha_{0}=0.5,\xi =1.1$. The optimization process is terminated when the updated value of the path parameter is smaller than the user-specified positive value $\left|{x^{*}}_{k+1}-x_{k}\right|\leq 0.65$.

To verify the convergence of the algorithm, three different groups of the path parameters $(r_{0}=49$ m, $v_{0}=16\;\mathrm{kn})$, $(r_{0}=39\;m,v_{0}=12\;\mathrm{kn})$, and $(r_{0}=28\;m,v_{0}=10\;\mathrm{kn})$, are chosen as the initial values for the IPT simulations. To compare the performance of the TPA-IPT method proposed in this study with the fixed step-size version in [28], simulations using both methods are conducted for each group, and the results are shown in Figure 9 and Table 3. Figure 9a,c,e show the convergence of the cost function value and the two path parameters for each initial value set using TPA-IPT, and Figure 9b,d,f are their counterparts for the fixed step IPT. It is observed that both algorithms can achieve a similar minimum point with small variance (Table 3) regardless of how the initial value changes. However, the convergence speed of the TPA-IPT is much faster than that of the fixed step IPT.

To perform a better comparison, the descent processes of the cost function value are drawn in Figure 10 for the first group of the initial value by using both a contour graph and in a 3D cost function surface. The sharp slope of the function indicates that TPA-IPT speeds up and arrives at the minimum point with only 19 iterations, which is approximately one third of the number of iterations needed using fixed step IPT. For each iteration, TPA-IPT needs 2 more trials to adjust the step length. Therefore, the total number of trials needed by TPA-IPT is 114, which is less than half of the trial numbers for the fixed step strategy. This difference helps decrease the working load of the IPT tests when it is applied in real vessel tests.

A similar result is obtained for other initial value setups. The variance of the step length using TPA-IPT is shown in Figure 11, and a drop in the step size at the last several iterations is observed. This result is very reasonable considering the need for the fine tuning of the parameters when they are close to the minimum point.

### 5.2. Path-Smoothing and Speed-Planning Simulations

Section 5.1 shows three sets of the initial path parameter values that are tested using the TPA-IPT method; these values provide close to optimal solutions for the turning radius and turning speed. The optimized tuning parameters $r_{r}=29.60$ m, $v_{r}=7.05$ kn will be used in the following work on the path-smoothing and speed profile-planning. To verify the effect of the path-smoothing and speed-planning algorithm, various turning conditions are designed. Four tested scenes, including a path with a large turning angle, a path with a small turning angle, a path for water sampling, and a path for bathymetric measurement, are designed to verify the performance of the proposed algorithm in this paper. The code was run on a computer with an Intel i5 2.30 GHz processor and 4.00 GB of RAM. To verify the real-time performance of the proposed algorithms, each test scene was run 10 times. As shown in Table 4, the average time used for the path and speed profile calculation in the four test scenes are 0.044 s/km, 0.072 s/km, 0.016 s/km and 0.079 s/km respectively. In addition, the path-following performances are listed in Table 5 with a comparison between the original and optimized path and speed profiles. As expected, the path-following accuracy and the drive safety are considerably improved when the method described in this study is used. The details on the results will be further discussed for individual cases in the following paragraphs.

Figure 12 shows the simulation results for the large-angle turning scene. Black curves are the planned path, blue curves are the high-speed USV trajectories, red curves are the low-speed trajectories, the purple curve is the acceleration trajectory, and the green curve is the deceleration trajectory. The corresponding speed change for Figure 12b,d are shown in Figure 13. When the USV makes a large-angle turn at a speed of 12 kn, due to the limitation of dynamics, a large tracking error will be produced; this error will oscillate when returning to track the straight line, as shown in Figure 12a,c. The maximum path error is 35.87 m, as shown in Table 5. After applying the path-smoothing and speed-planning work proposed in this paper, it was found that the USV slows down near the turning corner and follows the curve at a speed of 7.05 kn. This algorithm agrees with the dynamics of USVs; therefore, the tracking error is small, and the oscillation is considerably reduced, as show in Figure 12b,d. The maximum path error is 5.34 m, and the error value is insensitive to the path conditions. The maximum path error is reduced by 85%, and the driving of the USV is considerably stabilized with a small increase (13%) in the travelling time.

For tasks with less demands on small path deviation and turning safety but more on small travelling time, the weight factors in (Equation 1) should be readjust to have large $\lambda_{2}$ and small $\lambda_{1}$. In this case, the resulted turning radius would be small, and the turning speed would be large using the IPT method. However, the authors do not recommend high-speed turning considering the risk of capsizing.

When the USV turns at small angles, the path-following performances of the USV without and with using the method described in this paper are compared. The influence of its dynamic constraints will decrease, as shown in Figure 14a. Therefore, the maximum path deviation is originally 19.47 m, which is smaller than that for the large-angle case. By applying the path-smoothing and speed-planning algorithm, the maximum path deviation is decreased to 5.06 m, as shown in Figure 14b. The displacement of the optimized path is 1.3% less than that of the original path, and the time used is 15.5% more than that of the original path. With a slight sacrifice of travelling speed, the method described in this paper shortens the travelling distance and, more importantly, considerably decreases the path-following error.

The decreased path deviation sometimes is vital for driving safety. As shown in Figure 14a, the USV crashed onto an obstacle using the original path and speed profiles when it is turning into a narrow passage. The optimized path and speed design largely decreased the path deviation, and the USV successfully passed through the cluttered area, as shown in Figure 14b.

Various situations for speed profile-planning are also designed to verify the algorithm in Section 4. As shown in Figure 14, the line segment between the triangular obstacle and the rectangular obstacle is too short to achieve full acceleration, which corresponds to the condition 2 in Section 4 for the speed profile-planning. With the acceleration and deceleration tests in Figure 15a, the corresponding displacement curves are calculated and shown in Figure 15b. Thus, $d_{a}=172.04$ m, $d_{d}=138.62$ m, $d_{\min}=310.66$ m. The length of the second line segment in Figure 14d=198.97 m <310.66 m. Thus, (Equation 14) is used to plan the speed profile for the segment. The acceleration and deceleration distances in the straight-line segment can be obtained as $d_{1}=73.5$ m, $d_{2}=125.47$ m. As shown in Figure 16, the USV can reduce the speed to the optimal value before turning at the next corner.

A similar result is obtained for the water sampling task around an island. A map of Mount Putuo, which is one of the 1390 islands in Zhoushan Archipelago, China, is drawn in Figure 17. The sampling points scatter around the island and the USV needs to visit these points accurately to retrieve the water samples.

During the mission, the USV encountered two small-angle turns and six large-angle turns. As shown in Figure 17a, the maximum path error at the corners by the original path and speed setup is 34.96 m, so the USV largely deviates from the desired sampling locations. By smoothing the path and planning the speed, the maximum tracking error can be reduced to 5.53 m, as shown in Figure 17b. The motion stability of the USV is considerably improved, and the water samples are retrieved close to the desired locations.

The bathymetric measurement is another example to show the benefits of the IPT method. As shown in Figure 18, the USV needs to traverse an area to obtain the water depth data following a pre-defined path. This task requires the USV to have higher path-following accuracy and lower cruise speed [2]. The new weights of the cost function and the minimum turning radius in the IPT algorithm are reset as $\left(\lambda_{1},\lambda_{2},\lambda_{3}\right)=\left(5,4,278\right)$ and $r_{\min}=14$ m. The remaining parameters remain unchanged. The original parameters $r_{0}=39$ m, $v_{0}=12$ kn were optimized by IPT to be $r_{r}=16.66$ m, $v_{r}=4.96$ kn. According to the line spacing in Reference [38] and the property of the USV model, the line spacing is selected as 40 m.

As shown in Figure 18, the maximum path deviation generated by the USV using the original cruise speed is 30.40 m, which is larger than the line spacing and causes intersections between the paths at the corners. This could leave some places uncovered at the corner areas. The path-following performance is greatly improved after path-smoothing and speed-planning, and its maximum path deviation is 5.78 m. In addition, the smooth actions at the corners can also help to reduce the measurement noises caused by highly dynamical turning behaviors of the USVs.

## 6. Conclusions

This study proposes a practical and real-time solution for the path and speed profile generation by considering the influence of vehicle dynamics on hydrographic survey tasks. The offline learning method uses experimental data to obtain optimal turning radius and speed for the vehicle, which provides design constraints for the path and speed profile optimization. The improved IPT method is proven to be effective in obtaining satisfactory turning parameters within a limited number of trials, which balances the demands of small turning radius, fast turning speed and small path-following error. The resulting radius is used to smooth the sharp corners with the L-3C-L /L-2C-L path-smoothing method, which guarantees relatively small path deviations from the original path. The resulting turning speed is used to plan the speed trajectory and shape smooth transients between the fast-straight-line following mode and the relatively slow turning mode.

The simulation results show that the proposed path-speed profile optimization approach produces smaller path deviations and is more suitable for hydrographic survey activities. The relatively slow turning speed and smoothed path largely improves the path-following performance. Compared to the case using original path and speed profiles, the maximum path deviations decreased by 81.07% on average at a very small cost of time increase by 16.70%. The smoothness of the trajectory and the driving safeness are largely improved as well, and the calculation cost is low, which makes it potentially beneficial for real-time tasks in complex marine environments.

In addition, the algorithm proposed in this paper can be extended to unmanned systems with similar dynamical properties and demands on trajectories. For example, it can be used to optimize the path and speed profiles for an UAV system to have smoothed turning behaviors and more accurate path-following performance.

## Figures and Tables

**Figure 1 sensors-20-00439-f001:**
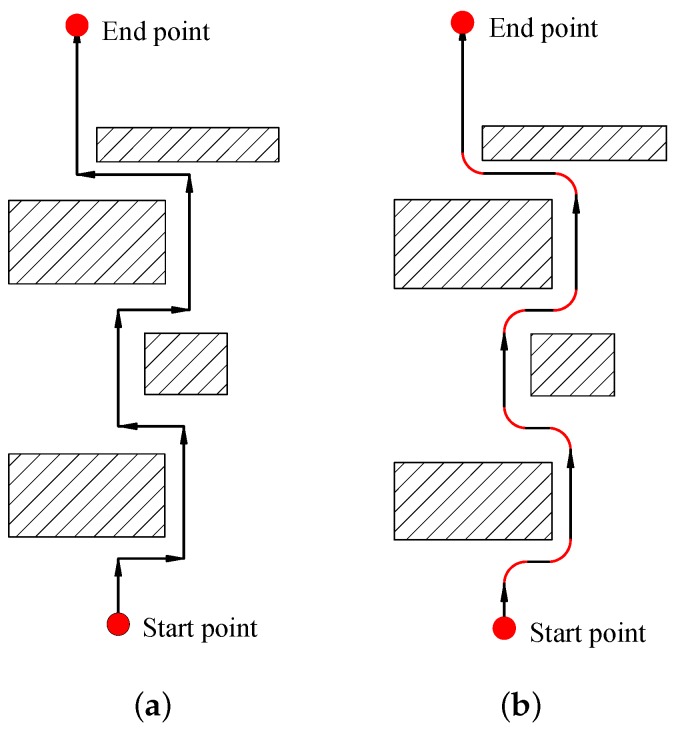
Original path and smooth path profiles: (**a**) Original path. (**b**) Smooth path.

**Figure 2 sensors-20-00439-f002:**
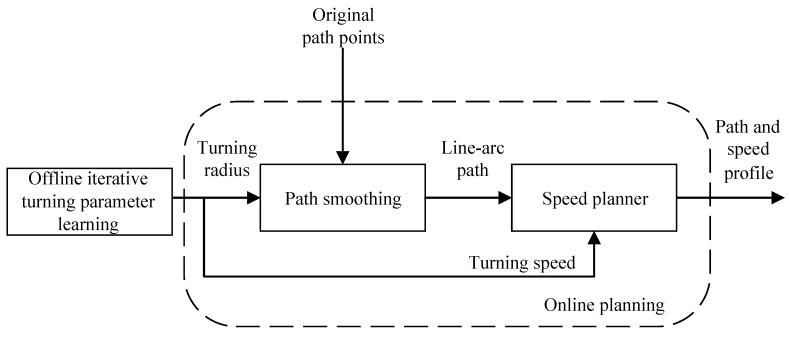
Path and speed-planning scheme.

**Figure 3 sensors-20-00439-f003:**
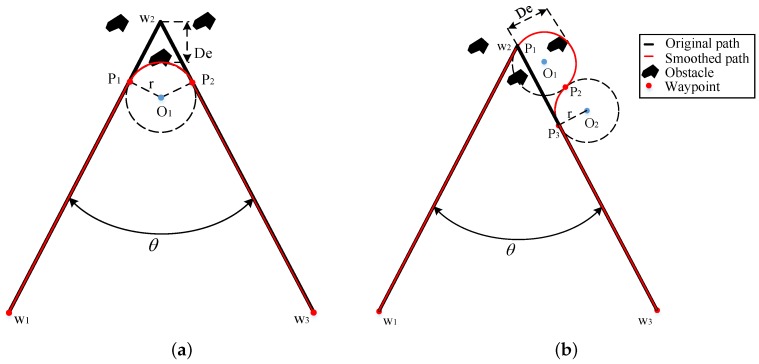
Smoothed paths: (**a**) Created by using one Dubins curve. (**b**) Created by using two Dubins curves.

**Figure 4 sensors-20-00439-f004:**
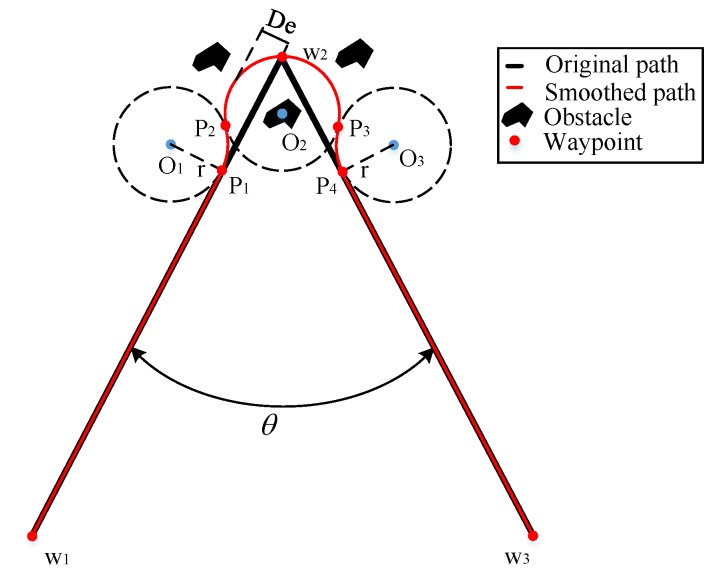
Smoothed path created using the L-3C-L method.

**Figure 5 sensors-20-00439-f005:**
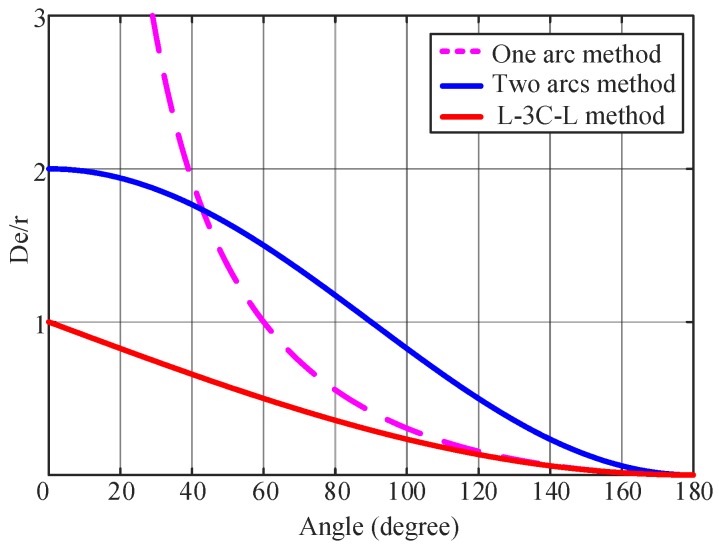
Path deviation comparison of the three smoothing methods.

**Figure 6 sensors-20-00439-f006:**
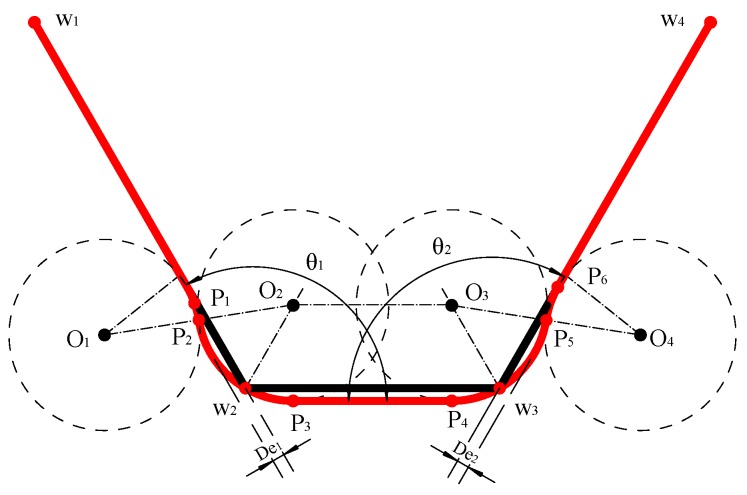
Adaptation to the special case with two close corners.

**Figure 7 sensors-20-00439-f007:**
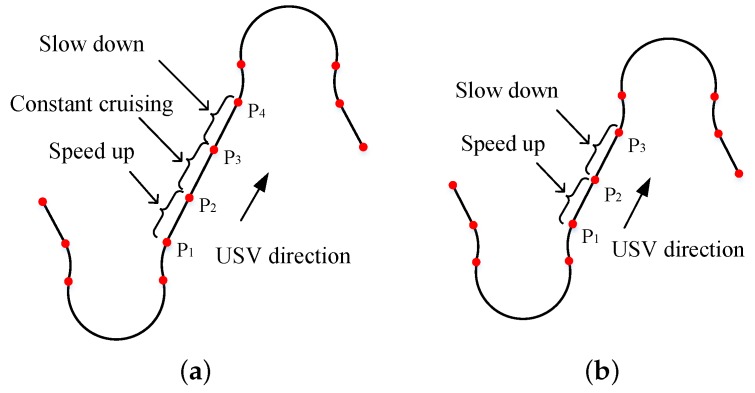
Speed design diagram: (**a**) The straight-line segment is sufficiently long. (**b**) The straight-line segment is not sufficiently long.

**Figure 8 sensors-20-00439-f008:**
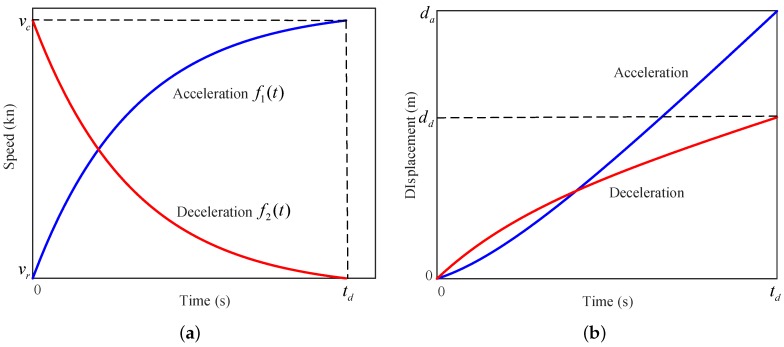
Acceleration and deceleration test: (**a**) Speed curves. (**b**) Displacement curves.

**Figure 9 sensors-20-00439-f009:**
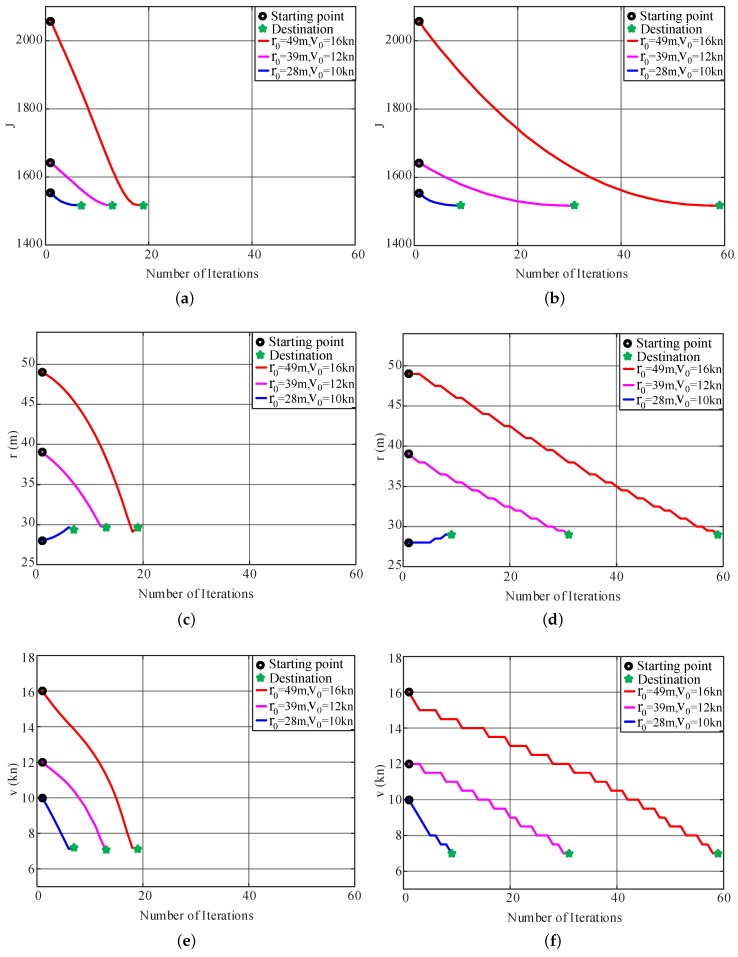
Optimization result graphs: (**a**,**c**,**e**) are the convergence curves of the cost function value and the two path parameters for each initial value set using TPA-IPT; (**b**,**d**,**f**) are their counterparts for the fixed step IPT.

**Figure 10 sensors-20-00439-f010:**
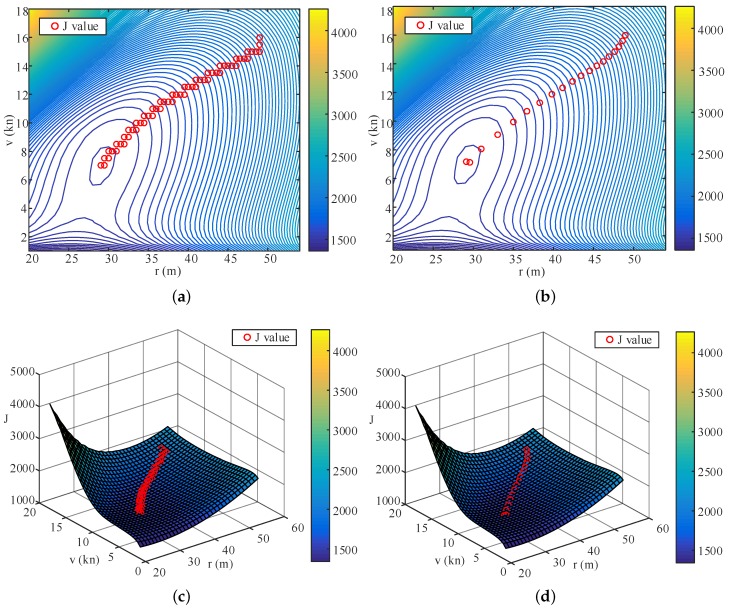
Diagram of descending processes with the first group initial values: (**a**,**c**) are the iterative processes of TPA-IPT, and (**b**,**d**) are the corresponding results of the normal IPT.

**Figure 11 sensors-20-00439-f011:**
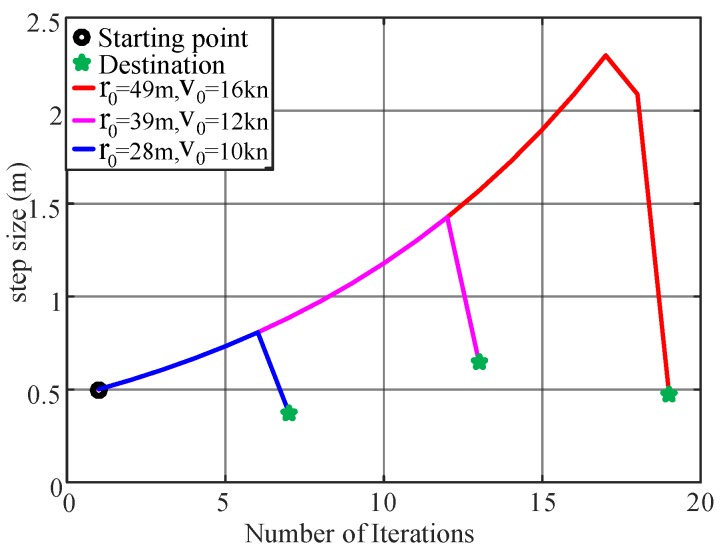
Step-size variations of the optimization process.

**Figure 12 sensors-20-00439-f012:**
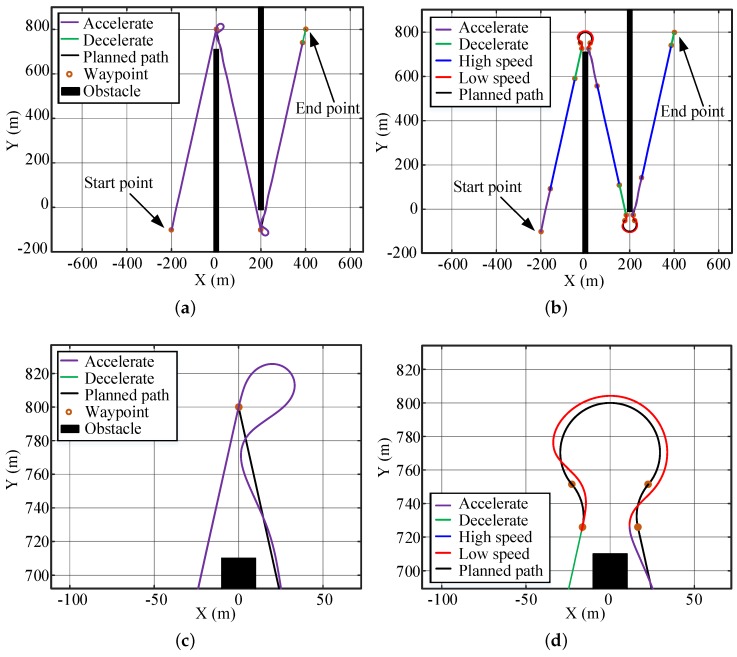
Simulation of large-angle turning tracking scene: (**a**) Original path-tracing results. (**b**) Optimized path-tracing results. (**c**,**d**) Enlarged partial views of the paths for the turns in (**a**,**b**).

**Figure 13 sensors-20-00439-f013:**
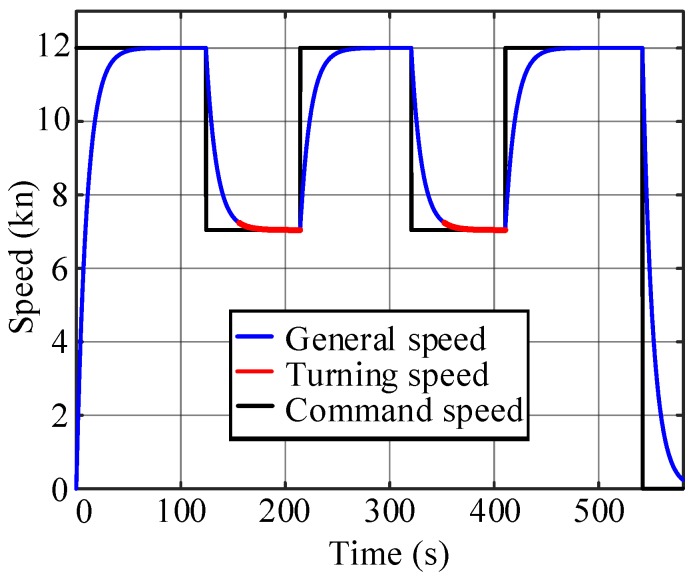
Speed curve of the large-angle turning tracking scene.

**Figure 14 sensors-20-00439-f014:**
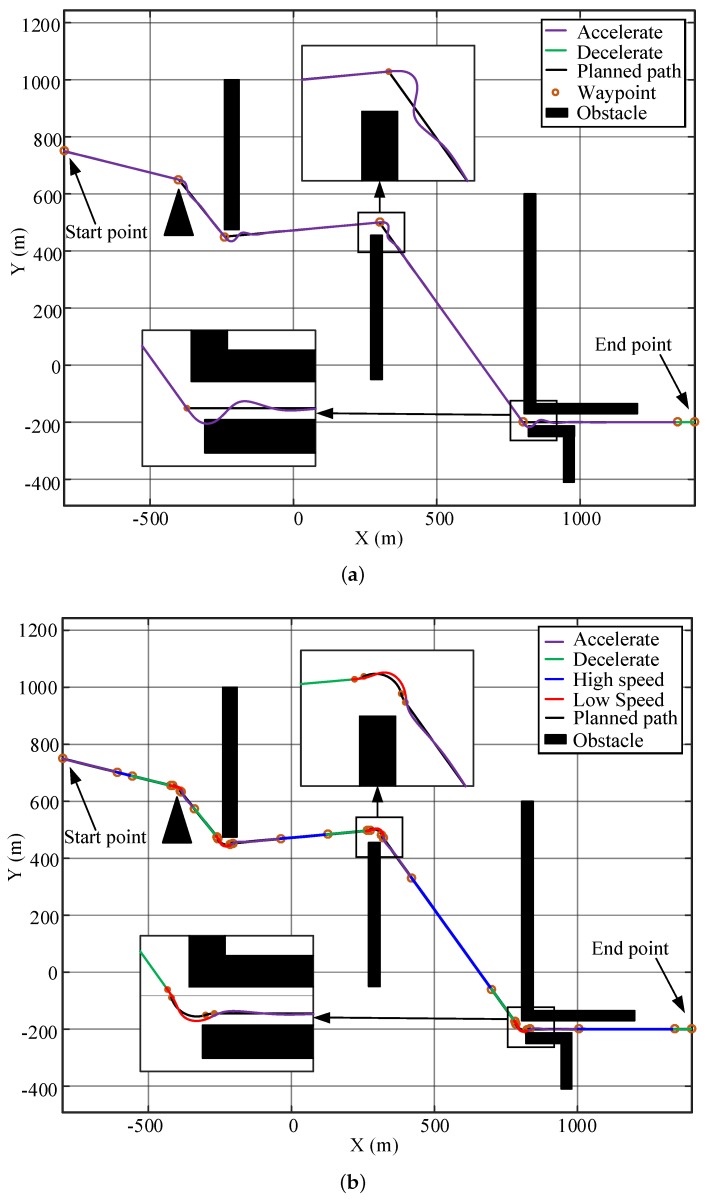
Simulation of the small-angle turning tracking scene: (**a**) Original path-tracing results. (**b**) Optimized path-tracing results.

**Figure 15 sensors-20-00439-f015:**
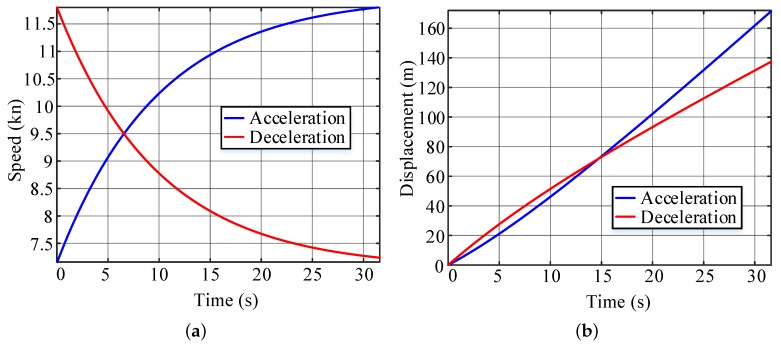
Acceleration and deceleration tests: (**a**) Speed test curves. (**b**) Displacement test curves.

**Figure 16 sensors-20-00439-f016:**
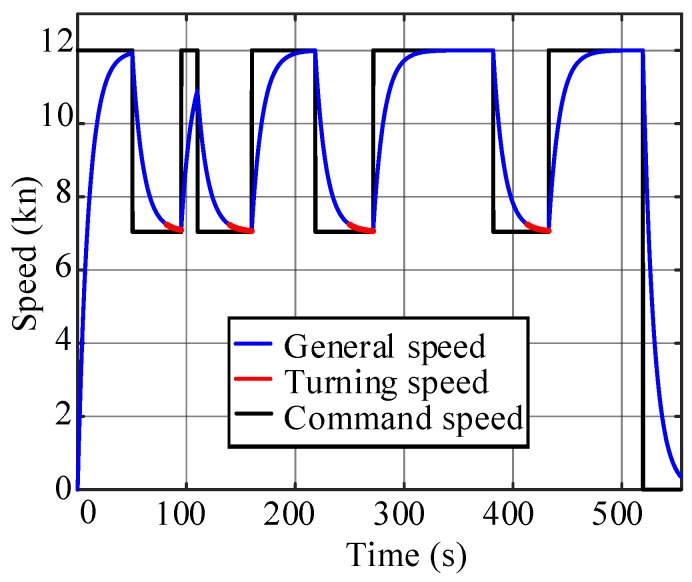
Speed curve of the small-angle turning tracking scene.

**Figure 17 sensors-20-00439-f017:**
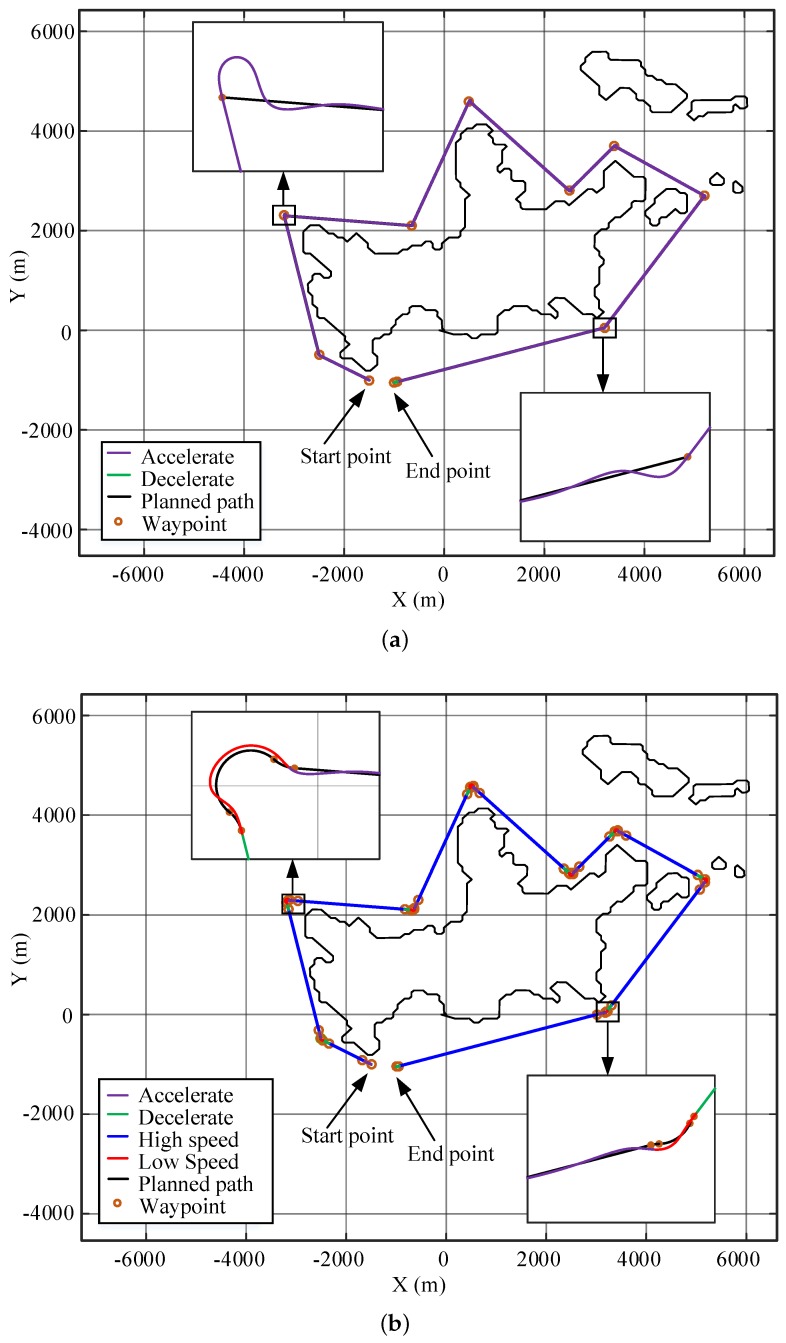
Simulation of water sampling: (**a**) Original path-tracing results. (**b**) Optimized path-tracing results.

**Figure 18 sensors-20-00439-f018:**
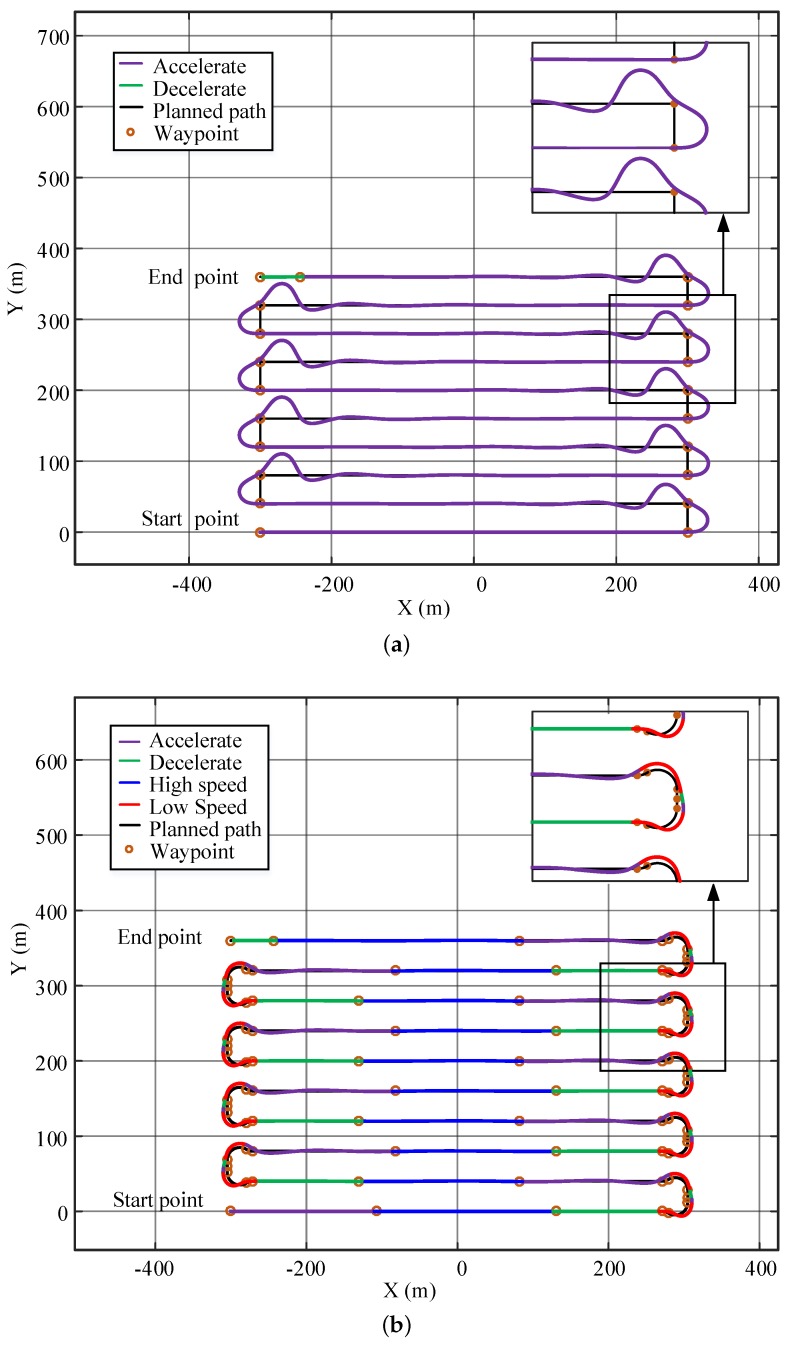
Simulation of bathymetric measurement: (**a**) Original path-tracing results. (**b**) Optimized path-tracing results.

**Table 1 sensors-20-00439-t001:** Vessel properties and controller parameters.

Vessel Properties	Value	Controller Parameters	Value
Long	8.5 m	$K_{p,u}$	0.1 1/s
Width	3.0 m	$K_{p,\psi }$	5.0 1/s
Mass	3980 kg	$K_{d,\psi }$	1.0 *s*
Inertia	19,703 kg/m${}^{2}$		

**Table 2 sensors-20-00439-t002:** Vessel Model Parameters.

Parameters	Value	Parameters	Value
$X_{u}$	−50	$X_{\left|u\right|u}$	−135
$Y_{v}$	−200	$Y_{\left|v\right|v}$	−2000
$Y_{r}$	0	$N_{\left|r\right|r}$	0
$N_{v}$	0	$X_{uuu}$	0
$N_{r}$	−3224	$Y_{vvv}$	0
$F_{x,\min}$	−6550 N	$N_{rrr}$	−3224
$F_{x,\max}$	13,100 N	$X_{u^{‘}}$	0
$F_{y,\min}$	−645 N	$Y_{v^{‘}}$	0
$F_{y,\max}$	645 N	$Y_{r^{‘}}$	0
$l_{r}$	4.0 m	$N_{v^{‘}}$	0
		$N_{r^{‘}}$	0

**Table 3 sensors-20-00439-t003:** Comparison of results with and without an adaptive step length.

	Radius (m)	Speed (kn)	Average Tracking Error (m)	Number of Iterations
Initial value	49	16	4.33	
Normal IPT	29.00	7.00	3.84	59
TPA-IPT	29.58	7.10	3.82	19
Initial value	39	12	4.21	
Normal IPT	29.00	7.00	3.84	31
TPA-IPT	29.60	7.05	3.81	13
Initial value	28	10	4.66	
Normal IPT	29.00	7.00	3.84	9
TPA-IPT	29.31	7.21	3.87	7

**Table 4 sensors-20-00439-t004:** Average runtime of algorithm.

	Large Angle	Small Angle	Water Sampling	Bathymetry
Average runtime (s/km)	0.044	0.072	0.016	0.079

**Table 5 sensors-20-00439-t005:** Tracking parameters of USV with and without route optimization.

Tracking Parameters	Large Angle	Small Angle	Water Sampling	Bathymetry
Original displacement (m)	2931.05	2728.44	23,265.18	6975.18
Optimized displacement (m)	2881.94	2693.33	23,123.05	6563.18
Original travelling time (s)	512.92	479.54	3811.95	1156.29
Optimized travelling time (s)	580.45	553.85	3987.13	1543.95
Original maximum path error (m)	35.87	19.47	34.96	30.40
Optimized maximum path error (m)	5.34	5.06	5.53	5.78

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
