# Peer review of "Iterative Learning-Based Path and Speed Profile Optimization for an Unmanned Surface Vehicle"

_sensors, 2020, doi:10.3390/s20020439_

Round 1

Reviewer 1 Report

This paper presents a path smoothing method by optimizing path parameters for USVs. The IPT method was used for optimizing the path parameters and the Dubins curves were adopted to smooth the path. Overall, the manuscript is well-structured and clearly written. There are several minor issues that should be corrected.

1) Line 102: p_i is the actual~ --> p_bar_i

2) Section 2.2: The reference for the line search method should be added although the method is well known.

3) The effects of lambdas in Equation (1) should be discussed in simulation results.

4) Writing a pseudocode of the learning method for path parameter tuning is recommended for better understanding.

Reviewer 2 Report

This paper presents a very interesting study concerning the application of a path planning algorithm for a USV.  There are some minor issues to address:

Equation (3) seems to be based on Euler integration.  Is this the case and does if affect the stability of the algorithm?  If so then how can this be improved?  Comment on this. Section 3 needs some references and could be condensed as it appears to contain a lot of theory that is not original.  Try to emphasize the originality. In Section 5 you should describe the model (provide equations with full model in appendix) and less on the computer/software used.  Explain why you are using a low fidelity model.  Do you perform the studies using a high fidelity model?  How does the model used for this study compare with the actual system? You should include a description of the control architecture used, the actuator configuration and provide more details about how this affects the performance of your algorithm. Some data plots seem to end before the end of the graph axes.  This does not look professional and should be avoided where possible (e.g. Figures 13, 15 and 16). Analysis of results needs to be extended, Conclusions need to be expanded to provide overall conclusion of the work. There are grammatical and spelling errors (e.g. don't start a sentence with "Because"). Avoid using the term "unit matrix" as it can be ambiguous.  Use "identity matrix" to avoid ambiguity. Some typos in the references.

Reviewer 3 Report

This is an excellent simulation paper proposing a combination of two-point step size adaptation with iterative parameter tuning for local path planning for navigating discontinuities in global paths for unmanned surface vehicles (USV). 

The paper does an excellent job presenting the general challenges in path planning and the approach developed by the authors. The proposed methods are then demonstrated with multiple simulations. 

The paper is well written, clear, and interesting to read. 

I would like to provide some specific notes on minor revisions that I feel would improve the paper. 

Line 19: greedy mechanism-based

Line 31: obstacle avoidance

Line 23: considering the kinematics

Line 37: Revise 'this deviation may fail the obstacle avoidance'. The deviation causes the vehicle to fail.

Line 42: Recommend a citation regarding the Dubins curve being the most widely used profile. 

Line 44: computationally cost efficient

Line 55: For speed planning

Line 61: what do other studies mean by driving safety? Is obstacle avoidance not driving safety? It may be helpful for the authors to be more specific about other studies' objectives.

Line 66: may increase 

Line 85: application in determining

Section 2 - The authors may feel that it is already clear but I would recommend that the authors include some text that adds more clarity to the authors choices regarding optimizing performance - the choice to optimize path deviations over other objectives (particularly speed/travel time).

Line 104: difference between the reference speed

Line 106: There appears to be extra space after subscripts throughout the document

Line 122: Despite it being spelled out in the header for 2.2, I would prefer to see Two-Point Step Size Adaptation spelled out here as well.

Line 127: robust enough

Line 152: content is an odd choice of words here. 

Line 153: target point

Figure 5: I would recommend a different color 3 - the reds are too similar. Possibly use dashed line patterns.

Line 203: deceleration section dd (subscript appears to be incorrect)

I would suggest that the authors review the discussion of path length for accel/decel and figure 8b to ensure that it is accurate and clear. 

Table 3. Vertical space before each set of initial values would help visually separate the three tests.

Figure 9. The different x axes obscure the differences in # of iterations. 

Line 225: odd spacing on the parentheses and commas

Line 247: provide close to optimal

Table 4: Add 'travelling' to Original time and Optimal time for clarity. I suggest that the authors add a brief discussion of tradeoffs between path deviation and travel speed. 

The paths are optimal based on the path deviation constraint. In the given cases, the original path does not generate a failure. Are these good demonstrations for the algorithm? Do the authors have an example that shows that the improved algorithm avoids a collision? 

Line 278: To me, it seems like some text is missing bridging line 277 to 278 where we transition from a description of path deviation improvements to the accel/decel behavior.

Also, is it the case that the line segment may not be sufficiently long or is it known to be too short?

Line 290: large-angle scenes? Do you mean large angle scene to be different than a large angle turn?

Line 294: improves the mission performance -- by what metric? Not travel time. This should probably be specific to path deviation. 

Line 300: The resulting radius

Details on runtime - the paper repeatedly mentions real-time solutions and computational cost but I don't see details on computational performance of the algorithms

Is any aspect of these algorithms specific to USV? 
